# The Understanding and Experiences of Living with Dementia in Chinese New Zealanders

**DOI:** 10.3390/ijerph19031280

**Published:** 2022-01-24

**Authors:** Gary Cheung, April Yuehan Su, Karen Wu, Blake Yue, Susan Yates, Adrian Martinez Ruiz, Rita Krishnamurthi, Sarah Cullum

**Affiliations:** 1Department of Psychological Medicine, University of Auckland, Auckland 1023, New Zealand; aprilyuehan@icloud.com (A.Y.S.); susan.yates@auckland.ac.nz (S.Y.); a.martinez@auckland.ac.nz (A.M.R.); sarah.cullum@auckland.ac.nz (S.C.); 2Mental Health Services for Older People, Counties Manukau Health, Auckland 2025, New Zealand; Karen.Wu@middlemore.co.nz; 3School of Management, Massey University, Auckland 0745, New Zealand; B.Yue@massey.ac.nz; 4Department of Demographics Epidemiology and Social Determinants, National Institute of Geriatrics of Mexico, Mexico City 10200, Mexico; 5National Institute for Stroke and Applied Neurosciences, Auckland University of Technology, Auckland 0627, New Zealand; rita.krishnamurthi@aut.ac.nz

**Keywords:** dementia, Chinese, health literacy, lived experience, coping

## Abstract

Little is known about the lived experience of dementia in the New Zealand Chinese community. This study aims to explore the understanding and experiences of living with dementia in Chinese New Zealanders. Participants were recruited from a memory service and a community dementia day programme. In-depth interviews were conducted by bilingual and bicultural researchers. The recorded interviews were transcribed and thematically analysed. Sixteen people living with dementia and family carers participated in this study. The first theme revealed the lack of understanding of dementia prior to diagnosis, the commonly used term of “brain shrinkage” and that dementia is associated with getting older. The second theme covered the symptoms experienced by people with dementia and how family carers found anhedonia and apathy particularly concerning. The third theme highlighted the tension between cultural obligation and carer stress. The fourth theme is about the stigma attached to dementia. Our results provide some insight into ways to improve dementia care for Chinese New Zealanders, including targeted psychoeducation in the Chinese community to improve awareness and to reduce stigma, access to person-centred interventions, and learning about strategies for healthy ageing to live well with dementia, and emotional support and psychoeducation for family carers to reduce carer stress.

## 1. Introduction

Dementia is recognised as a global public health priority by the World Health Organization (WHO) and Alzheimer’s Disease International (ADI) [1]. There are currently 50 million people living with dementia worldwide, and this number is projected to increase to 82 million in 2030 and 152 million in 2050 [2]. Dementia is a neurodegenerative disease that affects a person’s memory, thinking, behaviour and day-to-day functioning. A person with dementia often requires support from their family and friends to maximise their independence and wellbeing while living in the community. Therefore, well-informed services for both the person living with dementia (PLWD) and their carers are essential.

New Zealand is a bicultural country with many other ethnic minority groups, including European 70.2%, Māori 16.5%, Asian 15.1% and Pacific peoples 8.1% [3]. There is evidence that populations of older Māori, Asian and Pacific people are increasing at a faster rate than their New Zealand European counterparts [4], and a significant rise in the prevalence of dementia in these groups has been predicted [5]. A number of recent studies have explored the lived experience of dementia in New Zealand European, Māori and Pacific people. For example, a qualitative study with 49 mainly New Zealand Europeans living with dementia and their carers suggested that PLWD value their independence and are living meaningful lives, although they also reported that living positively with dementia is not easy [6]. Another study using focus groups of Māori elders and interviews of Māori families living with dementia highlighted the importance of spiritual wellbeing along with honouring their identity as kaumātua (Māori elders) [7]. They also concluded that Māori families are generally inclusive and have a strong obligation to care for others, which is crucial for the care of older Māori with dementia. Fakahau, Faeamani and Maka (2019) interviewed 20 Pacific families (nine Tongan, six Samoan, three from the Cook Islands, one Fijian, and one Niuean) living with dementia in Auckland, which has the largest number of Pacific Island people living anywhere in the world [8]. They found Pacific families often have a limited understanding of dementia. There is also a lack of Pacific health services providing dementia support services, and families are reluctant to admit their loved ones to nursing home care because the quality of care is perceived as a concern.

A previous study conducted with a group of Asian healthcare professionals concluded that much work is needed to destigmatise dementia in New Zealand Asian communities through psychoeducation, public awareness and the availability of readily accessible services that can meet their cultural and language needs [9]. The Chinese population is the largest Asian ethnic group in New Zealand (4.9% of total population) [3]. However, little is known about the lived experience of dementia in the New Zealand Chinese community and how this affects the lives of PLWD and their families. It is likely that different psychosocial and cultural issues are at play for Chinese PLWD in New Zealand, where there were historical discrimination and maltreatment issues against Chinese from the late 1860s through the 1950s [10], recent waves of immigration from Hong Kong and China in the past three decades, and dementia services that were developed mainly for Europeans using a westernised model of care.

The aim of this study was to conduct semi-structured interviews with Chinese PLWD and/or their families to explore their understanding of dementia and experiences of living with dementia. We used qualitative methodology in this study because it lends itself more readily to explore these phenomena in-depth. We hope to use this new knowledge to develop better dementia services that are culturally appropriate and responsive to the needs of the growing number of Chinese PLWD and their families in New Zealand.

## 2. Materials and Methods

### 2.1. Participants and Setting

We used a convenience sampling strategy to recruit potential participants from a memory service of a large teaching hospital and a community dementia day programme in Auckland, New Zealand. We identified 22 Chinese people with a diagnosis of dementia in these two services. They and/or their families were initially approached by SC, an old age psychiatrist at the Counties Manukau District Health Board memory service, or by the clinical coordinator at the community dementia day programme. We kept our eligibility criteria broad to capture a wide range of perspectives from PLWD and/or their families. Participation was voluntary and written consent was gained from all study participants. The only exclusion criterion for PLWD was when a person lacked capacity to consent to participate in the study. A $50 voucher was provided to each participant at the completion of the semi-structured interview. Ethics approval was gained from the Health and Disability Ethics Committee (reference number 17/CEN/126/AM01).

### 2.2. Semi-Structured Interviews

A topic guide was constructed by the research team based on the dementia journey including recognition of cognitive symptoms, diagnosis and management of dementia (see Appendix A for the topic guide). The questions aimed to capture information from the perspectives of the PLWD and/or their families, including their understanding of dementia, their experience of the dementia diagnostic process, psychosocial adjustments and support services following the diagnosis, and current concerns and difficulties. Demographic data were collected regarding the participants’ age, gender, marital status, place of birth and number of years living in New Zealand.

The face-to-face interviews were conducted by three of the authors and five other trained bilingual and bicultural research assistants with a health background (nursing, social work, occupational therapy and clinical psychology). The interviews were conducted by one or two interviewers in the study participant’s home and in their first language, which included Mandarin Chinese, Cantonese Chinese and English. Our research assistants were provided with an interview manual which outlines the background of the study, research procedures, and instructions on the interviewing process. They attended five 2-h training workshops prior to the commencement of the study and one further workshop for peer-review of their interviewing experience. The training included an introduction to qualitative research, informed consent, discussion of the topic guide, an interviewing demonstration, role playing and practising using the topic guide, tips on managing difficult interviews (e.g., when an interviewee becomes emotional and tearful), and the process of transcribing interviews. The interviews were conducted between July 2018 and April 2019. They were audio recorded and transcribed verbatim by the research assistants. The transcripts were entered into NVivo 12 [11], a computer assisted qualitative data analysis program, for data organisation and coding.

### 2.3. Data Analysis

Interview transcripts were thematically analysed using a six-phase process described by Braun and Clarke [12]: familiarising with the data, generating initial codes, searching for themes, reviewing themes, defining and naming themes, and producing the report. Two Cantonese-speaking researchers and two Mandarin-speaking researchers were involved in analysing the data. Each of the transcripts was independently coded by two researchers. All four researchers reviewed and agreed on the resultant themes in an iterative manner over three meetings.

The four researchers who performed the thematic analysis are overseas-born Chinese. GC is an academic old age psychiatrist who has lived in New Zealand for over 30 years. KW is a senior psychiatric nurse who specialised in older people’s health and has lived in New Zealand for over 25 years. Both GC and KW were able to bring their understanding of the Chinese culture and clinical knowledge when analysing and interpreting the data. AS is a mental health social worker with a master’s degree and has lived in New Zealand for five years. She provided the social dimension when analysing and interpreting the data. BY has lived in New Zealand for three years and has a master’s degree in a non-health related area, which minimised the bias of introducing existing knowledge into the data analysis.

#### Data Reporting

Due to the richness of the data, we will focus on reporting the understanding of dementia and the impact of dementia on the lives of PWLD and their families in this paper. We will report our study participants’ experiences with healthcare services in a separate paper. When reporting the findings, we used pseudonyms for study participants in order to maintain confidentiality. We translated quotes from Chinese to English for publication purposes but illustrated some of the important quotes in Simplified Chinese characters to minimise any loss of meaning in translation.

## 3. Results

There were a total of 16 study participants, and Figure 1 shows the study participant flow: five dyads of a person with mild dementia (with capacity to consent to participate in the study) and a family member (husband: N = 1; wife: N = 1; son: N = 2), one person with mild dementia, and five family members of people with more severe dementia (wife: N = 2; daughter: N = 2; son: N = 1). A summary of their demographic information is shown in Table 1.

The average interview duration varied from 18 to 86 min in length (average 52 min). Four themes emerged from the thematic data analysis, and they are summarized in Table 2.

### 3.1. Theme 1: Understandings of Dementia

#### 3.1.1. Subtheme 1: Lack of Understanding of Dementia Prior to Diagnosis

Most participants had no knowledge or very limited knowledge of dementia prior to diagnosis. For example, Wei said: “What actually is dementia? I have never heard of it.” Jo “knew nothing” about dementia before her mother was formally diagnosed. She then researched about dementia on the internet and her knowledge of dementia improved “suddenly”. John, a well-educated man supporting his mother with dementia, had the following conversation with the research interviewer:


*Interviewer: Sorry John. So before your mother was diagnosed with Alzheimer’s dementia, did you have any like knowledge about dementia. What did you know about dementia?*

*John: Nothing, I was very ignorant until later on.*


This lack of awareness could have contributed to a delay in seeking help and diagnosis. For example, Yan had symptoms of dementia for three to four years before her husband took her to see their family doctor. Her husband did not understand the presentation of dementia and initially thought Yan’s symptoms were mood related.

#### 3.1.2. Subtheme 2: The Concept of “Brain Shrinkage”

Many of our PLWD and their family carers used the term “brain shrinkage” (“脑萎缩”) when describing dementia. Lan remembered her husband’s doctor described how the brain of an older person “shrinks”, as opposed to brain “growth” in children. Ming said: “There is shrinkage inside my brain.” Sandy also used “brain shrinkage” when describing her mother’s dementia, while Ying remembered that her husband’s specialist used the same term when he explained the dementia diagnosis.

With the “brain shrinkage” explanation, some family members gained considerable understanding of dementia. Sandy described that language, personality and emotion could be affected when the shrinkage progress of dementia involves certain parts of the brain. Similarly, Joe thought dementia can affect people differently; for example, some people may “wander” and some may have “moods”. Ying became aware of the main subtypes of dementia. She explained that one type of dementia is caused by the presence of a “protein” in the brain (Alzheimer’s disease) and the other type is caused by strokes (vascular dementia). Yong recently read on the internet that it is possible for hippocampi to re-grow and memory to improve. This gave him hope for his wife’s dementia. However, most carers believed that dementia is a progressive illness and that its symptoms will deteriorate over time.

#### 3.1.3. Subtheme 3: Dementia Is Part of Getting Older

Many of our participants considered dementia to be a part of growing older and the natural life cycle. For example, the phrase “laws of nature” (“自然规律”) was used separately by two family carers. Lan, who supports her husband, who lives in a nursing home, thought it is “very typical” for older people to have memory decline. Na touched on the topic of dying in old age while we were discussing dementia in older people, while Jing thought older people should accept the fact of death and dying. Our participants also suggested other medical and psychosocial factors that could play a role in cognitive functioning and the aetiology of dementia. Sandy noticed that her mother started to ask questions repeatedly after an operation and wondered whether anaesthetics may have side effects on the brain of an older person. John thought his mother’s memory gradually declined after she stopped babysitting his children and lost her role. Wei thought “tension” could make his thinking and memory worse; while Lan thought her husband’s poor temper and irritability could make him more susceptible to dementia. Lan also thought a good “mindset/mentality” may reduce the risk of dementia.

### 3.2. Theme 2: Impact on the PLWD

#### 3.2.1. Subtheme 1: Memory-Related Problems Are Common

Our study participants frequently described symptoms of short-term memory and forgetfulness in dementia: “always forgetting about things” [Yan]; “poor memory” [Wei]; “can’t remember things” [Ming]; “some memory decline” [Lan]; “memory is really poor” [John]. Wei, who has mild dementia and still works as an operator in a factory, recognized that he asks questions repeatedly because he does not want to make mistakes at work. However, his co-workers and family often respond: “I’ve told you already”. John found his mother’s repeated questioning stressful; for example, she asks John’s children their age 20 to 30 times during the course of a dinner.

#### 3.2.2. Subtheme 2: Anhedonia, Apathy, Other Neuropsychiatric Symptoms and Functional Impairment

A few PLWD and family carers found anhedonia and apathy particularly concerning. For example, Wei is no longer interested in playing mah-jong, while Sandy noticed her mother no longer enjoys social conversations and “has lost interest in everything”. Other participants with dementia had given up some of their previous social activities. For example, Na no longer goes to her church. Sandy said her parents do not go out very much and do not have many friends. Jo described it as “painful” for the family to see her mother who has lost interest in everything, but she wondered if her mother may not be suffering because of her apathy (“无欲无求”). Sandy also mentioned her mother’s apathy and believed that perhaps her mother is not aware of the changes.

Family carers also described a range of neuropsychiatric symptoms in dementia including language and communication difficulties (e.g., substituting people’s names, forgetting what to say next in a conversation, reduced verbal fluency and expression), visual hallucinations, misidentification, depression, irritability, mood lability and personality change. They reported that their family members were becoming more dependent with dementia. Their functional impairment included mobility and falls, instrumental activities of living (e.g., preparing meals, using the telephone, taking medication) and basic activities of daily living (e.g., personal care, bathing, eating). Jing described how her husband often gets angry and blames himself for not being able to function independently. A number of psychological impacts on the PLWD were also reported, including loneliness, low self-confidence and self-esteem. Jing thought people with dementia should be encouraged and not criticized for making mistakes due to their cognitive problems.

#### 3.2.3. Subtheme 3: Different Ways of Coping

Our participants with dementia use different mechanisms and strategies to cope with the cognitive impact of their illness. For example, Wei tends to normalize his memory problem: “There is no reason why memory would get better with age, in the same way that a machine will break as it ages”. Na writes everything down to remind herself of the things she needs to do. She also keeps to a routine and tries not to get worried. Wei thought physical exercise could help his memory, while Sam’s mother practices tai chi once a week with a small group of older people in the local park. Another PLWD practices calligraphy as a hobby and a couple of people with dementia like watching television at home.

### 3.3. Theme 3: Impact on Family Carers

#### 3.3.1. Subtheme 1: Cultural Obligation

There is general acceptance of the Chinese culture and the obligation that children are expected to take care of their elderly parents. “Of course older people are looked after by their children” [Sam]. John visits his mother every day to make sure she has taken her medication and eaten her meal. He “shares this responsibility” with his siblings because “we are Chinese”. Similarly, Fen is well supported by her seven children, and as her son Joe said: “… so we do rotate, we have a certain day, like my sisters sort of like the weekends….we cook for her and whatever, so we do whatever, she is well looked after.” Jing enjoys her children and grandchildren being around and she finds them very caring (“我们家孩子很孝顺”), while Jo thought she is blessed because she can still support her mother. However, with the global migration phenomenon, only one of Na’s four children lives in New Zealand and he visits her every week.

Two wife-carers reported a lack of support from their extended family because they do not have any children. Lan, whose husband is in a nursing home, lives alone and is the main support for her husband. She is not close to her brother who lives in the same city. Lan has a strong Christian faith and her main support comes from her church. She dedicates her life to her husband and visits him in the nursing home every day, to the extent that it has compromised her social activities. For example, her friends no longer invite her to their homes for dinner because they know she spends all day looking after her husband. Ying also visits her husband in the nursing home every day and helps with some of his personal care. She thought Chinese families are more caring than the other ethnic groups and her visits are important to maintain her husband’s wellness.

#### 3.3.2. Subtheme 2: Carer Stress and Coping

Caring for a family member with dementia is often associated with carer stress, and this is of no exception to our Chinese carers. Jo’s mother requires 24 h supervision. She finds caring for her mother “very tiring” and that it restricts her life. She does not leave her mother at home alone because she worries her mother may have an accident and she does not know how to use the telephone. Yong also finds caring for his wife “day and night” stressful, and he is tired. He makes sure she does not wander out of the house and is most affected by her mood swings. She often “hallucinates” and thinks people are accusing her and she can get irritable. Ming worries about not being able to find his way home if his wife does not accompany him to the supermarket. Similarly, Sam’s mother needs to be accompanied when going out.

John’s mother has both dementia and depression. He does not mind her repeated questioning but finds her depression more stressful. She previously attempted suicide by taking a medication overdose and often talks about dying. Sandy reported that her father, who used to be extremely good to her mother, is starting to lose patience (e.g., “yelling at her”) as her mother’s dementia progresses.

Our carers use a range of strategies to maintain their well-being. For example, engaging in leisure activities (e.g., singing, listening to music), physical exercises (e.g., swimming), socializing with friends, having strong religious faith, using humour, having a routine and “taking one day at a time”. Yong thought some family carers may not have the energy and resources to take care of their loved ones with dementia and that this burden could perhaps be shared by the community. Jing finds helping other older people in the community gives her a sense of being valued and improves her mood.

#### 3.3.3. Subtheme 3: Importance of Hope

Hope is an important coping mechanism used by many of our family carers who talked about their wish that there is a medical breakthrough for curing dementia. For example, Ying hopes there is a new medication that can remove amyloid plaques in the brain, while Sandy hopes that advances in research could find a cure for dementia and benefit other older people. Li thought her husband’s specialist should not have told him that his dementia has no cure but to give him some hope: “even though it is the reality, it is still meaningful to give him some hope” (“给他点盼望”). And indeed, Jo was unhappy when her mother’s family doctor told them that dementia has no treatment “except going to a nursing home”. However, acceptance and realization is essential for some carers to come to terms with the diagnosis; as Jo later said, “I need to accept faith and the diagnosis if we cannot change it.” Jo started to give up thinking her mother would improve six months after the diagnosis. Another carer said: “I have a feeling that there is no cure for my mother, but may be able to slow down the deterioration.”

### 3.4. Theme 4: Stigma

Both PLWD and family carers described the stigma attached to dementia, which affects the person with dementia themselves and the families of people with dementia. Yan believed that Chinese people generally do not like telling other people, including their family doctor, about dementia and mental health issues. Jing said that only adults in his family, not children, are informed about her husband’s dementia. She also does not want to tell people outside of her immediate family because she does not know how people would react or perceive her family. Lan has communicated to friends in her church that her husband has dementia and they have visited him in the nursing home.

Yong suggested the New Zealand government could provide Chinese people with more education on dementia. He believed the stigma of dementia in the Chinese community might prevent people from seeking information, diagnosis and support earlier. Li advocated the promotion of an age-friendly community. She thought the support, including emotional support, from the wider community could benefit the health of older people and the people who are providing the support.

## 4. Discussion

This is the first study ever conducted to explore the lived experience of dementia in Chinese people in New Zealand. The first theme revealed the understanding (or lack of) of dementia, the commonly used term of “brain shrinkage”, and that dementia is associated with getting older. The second theme covered the cognitive and non-cognitive symptoms of dementia experienced by PLWD and how their families found anhedonia and apathy particularly concerning. The third theme highlighted the tension between cultural obligation and carer stress and the importance of hope as a coping mechanism. The fourth theme is about self- and family-stigma attached to dementia.

Our participants commonly referred to the term “brain shrinkage” when they described dementia, and this finding requires further consideration. On reflection, the four researchers involved in the qualitative analysis believe the Chinese term of brain shrinkage “脑萎缩” has a relatively neutral connotation when the participants described it. And indeed, “brain shrinkage” provided a means for our participants to understand the neurodegenerative nature and medical cause of dementia. Although some of our participants considered dementia and memory decline as more prevalent in older age, none of our participants related dementia to mental health or a folk explanation [9]. This finding is consistent with the results of a study in Hong Kong where 13 out of 15 Chinese participants with dementia interpreted their illness as part of getting older, which might lead to delays in seeking help, diagnosis and treatment [13]. There has been much effort to address dementia-related stigma by changing the local names for dementia in East Asian Countries including China, Taiwan, and Singapore, where Chinese people are the majority residents [14]. Our participants seemed to accept the term “brain shrinkage”. It is possible that “brain shrinkage” provides a visual representation of an organ that is failing and that dementia is a brain disorder. This finding is consistent with a previous local study with a group of Asian healthcare professionals who suggested that Asian people are more accepting of medicalised problems than they are of those of a psychological nature [9]. However, we believe psychoeducation on strategies for healthy ageing (such as promoting social connectedness, physical exercises, and cognitive stimulation) would be useful for Chinese people who seem to just accept dementia as a medical problem or part of getting older and that the brain will continue to shrink.

It was most unexpected when a couple of elderly family carers mentioned amyloid plaques, hippocampi, and subtypes of dementia. Despite a generally low dementia literacy prior to diagnosis, our carers’ knowledge of dementia improved following the dementia diagnosis, possibly through health information available on the internet. The low dementia literacy prior to diagnosis is consistent with a recent online survey in China that used the Alzheimer’s Disease Knowledge Scale [15]. They found that dementia knowledge amongst their participants (the majority of them were highly educated and younger than 40 years old) was lower when compared to other high-income countries such as Germany, US and South Korea [15]. Their survey also found that 91% of the respondents answered correctly that a person with Alzheimer’s disease would have a gradually worsening ability to remember new information [15]; and we also found that our participants had good recognition of memory problems and functional impairment through their lived experience of dementia.

Our study identified that family carers felt uncomfortable, and sometimes distressed, by the non-cognitive symptom of anhedonia and apathy. Calia, Johnson and Cristea (2019) explored the cross-cultural representation of dementia and found that their participants in China, aged between 18 and 75 years, associated dementia with the following words in terms of high importance and high frequency: disease, memory loss, sadness, forgetfulness and Alzheimer’s [16]. It is interesting that “sadness” was found to be associated with dementia, which has a similar meaning to anhedonia and apathy reported by our family carers. This could lead to a danger of potentially treatable depression not being diagnosed in a PLWD. The use of person-centred interventions could be useful to address the cognitive and non-cognitive symptoms of dementia. For example, Cognitive Stimulation Therapy (CST), a cognitive intervention with the strongest evidence for improving cognition and quality of life, could be used for Chinese people with dementia living in the community [17,18]. CST is a group psychosocial treatment for people with mild to moderate dementia, and has been adapted for Chinese people [19]. There are 18 principles underpinning the CST programme and many of them are aligned with the concept of living well with dementia: person-centred, respect, involvement, inclusion, choice, maximising potential and building/strengthening relationships [20].

Our findings that Chinese families have an obligation to care for their elders with dementia and the practicing of filial piety are consistent with international literature [21,22,23,24,25]. Indeed, Chinese Canadian dementia carers felt taking care of their parents was more than a responsibility; “it was the way of life” [23]. They also lived with worries and emotional struggles and their needs were often shaped by their traditional beliefs [23]. A study of Singapore-Chinese dementia carers found that intergenerational connection and closeness are one of the important Chinese family values [25]. Previous Australian studies have suggested that Chinese communities prefer to provide dementia care at home (including day activities, community nursing and home help), rather than nursing home care [21,26]. Our finding of the tension between filial piety and carer stress in our Chinese family carers certainly echoed the experience of the physical and emotional work of caregiving that was closely intertwined with cultural issues in the Chinese-speaking community in Australia [21]. Dementia-related stigma is a common theme reported in previous international and local studies of dementia in Asian and Chinese immigrant communities [9,27,28], which would have added an extra barrier for PLWD and family carers to seek support to lessen the burden of caregiving. In terms of addressing carer stress and family stigma, there are a number of promising programmes developed for Chinese dementia carers, and they can be delivered in various modalities including face-to-face groups and via telephone and internet [23,29,30]. These programmes typically include a psychoeducation component and sessions aimed to provide emotional support for the carers and to encourage the use of formal social supports. For example, the Coping with Caregiving Group Program in Hong Kong uses cognitive behavioural strategies for Chinese carers in Hong Kong to gain self-efficacy for controlling their upsetting thoughts and handling the disruptive behaviours of the care recipients [29].

There are a number of limitations in this study. Firstly, the study participants were conveniently recruited from one location in New Zealand, and this could limit the generalisability of our results to other parts of New Zealand. However, nearly half (48.5%) of the Chinese population in New Zealand live in Auckland, where we conducted our study [3]. Secondly, our sample size was relatively small, although the qualitative data collected were rich and the findings that emerged from thematic analysis were consistent with international literature on this topic. For example, a general lack of awareness of dementia in the Asian community; dementia being seen as a common part of aging; dementia-related stigma; children feeling obligated to care for their parents and valuing filial piety or having respect for their elders [27,28,31,32,33]. Our research group has used the same qualitative study design to explore the understanding and experiences of living with dementia in some of the other main ethnic minority groups (Indian, Tongan and Samoan) in New Zealand. We will be able to compare the results of this study with the other ethnic minority group studies once they are all completed. Thirdly, we only included PLWD who had capacity to consent to participate in the study, and therefore our results are from the perspectives of people with mild dementia. We included family carers of a person with more severe dementia in an attempt to understand their experiences, but acknowledge that the perspective of people with more severe dementia is lacking. Fourthly, we decided not to objectively measure the characteristics of the PLWD (e.g., their follow up and involvement with the memory service or dementia day programme, cognitive function, and behavioural and psychological symptoms of dementia) because it could interfere with the qualitative interviewing process. However, we understand that the severity of cognitive impairment and BPSD could affect the experiences of carer stress [34]. Lastly, although the qualitative interviews and analysis were conducted in Chinese, we reported our main findings in English. The data analysis performed by the four bilingual and bicultural researchers provided a mean for peer reviewing our English report, but we acknowledge that nuances could have been missed, and meaning could have been added as a result of the translation process.

## 5. Conclusions

Our study findings have provided some insight into ways to improve dementia care for Chinese New Zealanders. Targeted psychoeducation in the Chinese community could improve the awareness of dementia and reduce the stigma attached to dementia. This could result in earlier recognition and diagnosis of dementia and better use of healthcare and formal support services. The “brain shrinkage” term provides a mean for Chinese people to understand dementia and accept it as a medical diagnosis. We would certainly like to explore the acceptability of this term further in future research with larger groups of Chinese lay people and healthcare professionals locally and internationally. Improved access to person-centred interventions and learning about strategies for healthy ageing would provide Chinese PLWD and their families the resources to live well with dementia. Chinese family carers would benefit from structured education and support programmes to help them with negotiating the conflict between filial piety and carer stress and to instil hope while caring for their loved ones as part of the dementia journey.

Since the completion of this study, the research team has partnered with two community organizations in a new project called “Caring for people with dementia together” (“護腦同行”), with the goal of achieving better outcomes for Chinese PLWD and their families.Using some of our study findings, the priorities of this new project are placed on increasing dementia awareness within the Chinese community, providing psychoeducation and support for Chinese family carers, and implementing a living well programme for Chinese PLWD.

## Figures and Tables

**Figure 1 ijerph-19-01280-f001:**
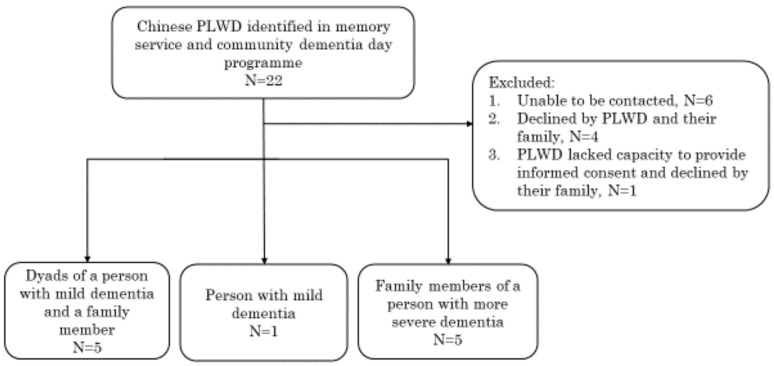
Flowchart of study participants. PLWD: person living with dementia.

**Table 1 ijerph-19-01280-t001:** Study participants and their demographic information.

	Pseudonym of Person with Dementia	Pseudonym of Family Carer (Relationship)	Age	Gender	Marital Status	Birthplace	Years Living in New Zealand	Language
Person with dementia	Wei	--	61	M	Married	Hong Kong	41	Cantonese
Dyad 1	Ming	--	87	M	Married	China	25	Mandarin
	--	Fang (wife)	83	F	Married	China	25	Mandarin
Dyad 2	Li	--	84	F	Married	China	20	Mandarin
		Jing (husband)	89	M	Married	China	20	Mandarin
Dyad 3	Yong	--	79	M	Married	China	19	Mandarin
	--	Yan (wife)	78	F	Married	China	19	Mandarin
Dyad 4	Na	--	75	F	Married	China	17	Mandarin
	--	Sam (son)	53	M	Married	China	22	Mandarin
Dyad 5	Fen	--	88	F	Widowed	China	81	English
	--	Joe (son)	60	M	Married	New Zealand	60	English
Carer 1	--	Ying (wife)	67	F	married	China	11	Mandarin
Carer 2	--	Lan (wife)	64	F	Married	Cambodia	20	Cantonese
Carer 3	--	John (son)	48	M	Married	Hong Kong	28	Cantonese
Carer 4	--	Jo (daughter)	49	F	Married	China	20	Mandarin
Carer 5	--	Sandy (daughter)	55	F	De facto	China	25	Mandarin

**Table 2 ijerph-19-01280-t002:** A summary of the qualitative analysis results and potential interventions to address these findings.

Main Themes	Subthemes	Potential Interventions
1. Understandings of dementia	(i) Lack of understanding of dementia prior to diagnosis (ii) The concept of “brain shrinkage” (iii) Dementia is part of getting older	Public awareness and psychoeducation on dementia and healthy ageing in Chinese community
2. Impact on the person living with dementia	(i) Memory-related problems are common (ii) Anhedonia, apathy, other neuropsychiatric symptoms and functional impairment (iii) Different ways of coping	Living well with dementia and person-centred evidence-based interventions e.g., Cognitive Stimulation Therapy
3. Impact on family carers	(i) Cultural obligation (ii) Carer stress and coping (iii) Importance of hope	Chinese carer psychoeducation and support groups
4. Stigma	-	Public awareness and stigmatisation campaign psychoeducation in the Chinese community

## Data Availability

Interview transcripts are available upon request.

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
