# Peer review of "The Understanding and Experiences of Living with Dementia in Chinese New Zealanders"

_ijerph, 2022, doi:10.3390/ijerph19031280_

Round 1

Reviewer 1 Report

In this qualitative study, the authors shared the results of investigating the characteristics of Chinese dementia patients residing in New Zealand. Dementia has a very large impact on the patients themselves and the caregiver who cares for them, and the difference occurs depending on the culture of each country. This study is interesting in that it provides basic data that can help understanding related to the characteristics of Chinese living in New Zealand. However, in order for the results of this study to be used in clinical practice, the following should be supplemented.

Minor comments:

  1. The authors noted that there are prior studies on dementia of other races residing in New Zealand. However, there is no data comparing the characteristics of Chinese with other racial characteristics identified in previous studies. The main purpose of this qualitative study is to reflect the differences in the characteristics of each race in the provision of treatment services in the future. Therefore, I ask that authors include the results of comparing the results of the previous study with the results of this study in the discussion.
  2. In addition, there should be an additional discussion on how to provide treatment services by utilizing the characteristics of Chinese people identified this time in the clinical field.

Author Response

In this qualitative study, the authors shared the results of investigating the characteristics of Chinese dementia patients residing in New Zealand. Dementia has a very large impact on the patients themselves and the caregiver who cares for them, and the difference occurs depending on the culture of each country. This study is interesting in that it provides basic data that can help understanding related to the characteristics of Chinese living in New Zealand. However, in order for the results of this study to be used in clinical practice, the following should be supplemented.

Minor comments:

  1. The authors noted that there are prior studies on dementia of other races residing in New Zealand. However, there is no data comparing the characteristics of Chinese with other racial characteristics identified in previous studies. The main purpose of this qualitative study is to reflect the differences in the characteristics of each race in the provision of treatment services in the future. Therefore, I ask that authors include the results of comparing the results of the previous study with the results of this study in the discussion.

Thank you for this suggestion. We cited two prior studies on dementia in other ethnic minority groups in New Zealand (Maori and Pacific peoples):

Dudley, M.; Menzies, O.; Elder, H.; Nathan, L.; Garrett, N.; Wilson, D., Mate wareware: understanding 'dementia' from a Māori perspective. N. Z. Med. J. 2019, 132, 66-74.

Fakahau, T.; Faeamani, G.; Maka, M. Pacific People and Dementia; Tongan Advisory Council: Auckland, New Zealand, 2019.

Different qualitative methodologies were used in these two prior studies. For example, Dudley et al. (2019) used focus groups instead of individual interviews to collect their qualitative data. Therefore, we do not want to compare our results directly with the results of these two prior studies.

The primary aim of our study was to conduct semi-structured interviews with Chinese PLWD and/or their families to explore their understanding of dementia and experiences of living with dementia (line 83). However, we have added the following in the Discussion (line 435):

Our research group has used the same qualitative study design to explore the understanding and experiences of living with dementia in other main ethnic minority groups (Indian, Tongan and Samoan) in New Zealand. We will be able to compare the results of this study with the other ethnic minority group studies once they are all completed.

  1. In addition, there should be an additional discussion on how to provide treatment services by utilizing the characteristics of Chinese people identified this time in the clinical field.

There is already some discussion on how to provide treatment services for Chinese with dementia based on the findings of our study (line 455-467). These potential interventions are now included in a new table (Table 2). We have also added the following to the Conclusion (line 468-474):

Since the completion of this study, the research team has partnered with two community organizations in a new project called “Caring for people with dementia together” (“護腦同行”) with the goal of achieving better outcomes for Chinese PLWD and their families.35 Using some of our study findings, the priorities of this new project are placed on increasing dementia awareness within the Chinese community, providing psychoeducation and support for Chinese family carers, and implementing a living well programme for Chinese PLWD.

Reviewer 2 Report

Please refer to the attached comments.

Author Response

The study entitled “The understanding and experiences of living with dementia in Chinese New Zealanders” conducted semi-structured interviews with Chinese person living with dementia (PLWD) and/or their families to explore their understanding and experiences of living with dementia in Chinese New Zealanders. Their results claimed to provide some insight into ways to improve dementia care for Chinese New Zealanders. Thank you for giving me the opportunity to review this paper. I have the following concerns regarding this investigation.

  1. Why did you choose this specific population to study? There were 3 groups in your study: Mandarin Chinese, Cantonese Chinese and English. Why not to include other populations speaking other Chinese dialects? Do you have any specific reasons?

This study is part of a larger project exploring the understanding and experiences of living with dementia amongst ethnic minorities in New Zealand. The main ethnic minority groups in New Zealand are Maori, Pacific peoples (mainly Tongan and Samoan) and Asians (mainly Chinese and Indian).

The following is added in the Discussion (line 435-437):

Our research group has used the same qualitative study design to explore the understanding and experiences of living with dementia in other main ethnic minority groups (Indian, Tongan and Samoan) in New Zealand.

We are not including Maori in our larger project because Dr Makarena Dudley, the lead author of the study “Dudley, M.; Menzies, O.; Elder, H.; Nathan, L.; Garrett, N.; Wilson, D., Mate wareware: understanding 'dementia' from a Māori perspective. N. Z. Med. J. 2019, 132, 66-74” (line 62), is part of our larger research group. Dudley et al. (2019) have already used Maori research methodology to explore the understanding of dementia in Maori.

We did not have a specific criterion to exclude populations speaking other Chinese dialects (line 96-98). Mandarin Chinese and Cantonese Chinese are the main Chinese dialects spoken by the Chinese population in New Zealand, and it just happened our Chinese speaking study participants spoke either Mandarin Chinese or Cantonese Chinese.

  1. Since the authors recruited potential participants from a memory service of a large teaching hospital and a community dementia day programme, they probably evaluated only patients with more advanced stages of the disease. What exactly are the inclusion and exclusion criteria for the study? A table describing all these with the ratio of compliance of participants (any withdrawn participants) is missing.

We kept our eligibility criteria broad to capture a wide range of perspectives from Chinese people with dementia and their families. The only exclusion criterion was when the person with dementia lacked capacity to consent to participate in the study. (line 96-99)

The suggestion that the memory service and dementia day programme evaluated only patients with more advanced stages of dementia is incorrect. The final sample included 6 participants with mild dementia and able to provide informed consent to participate in the study. There were 5 people with more severe dementia and lacked capacity to provide informed consent, but their family members were included in the study. We have added a new figure (Figure 1) to illustrate the flow of study participants.

  1. The authors did not describe the number of visits and evaluations (average) of their patients. Besides, how long each patient, on average, was followed up. There are many biases in the study.

These quantitative information are not absolutely essential for our qualitative study and would not have changed our thematic analysis and results. However, we have added your concern as a limitation (line 443-447).

Fourthly, we decided not to objectively measure the characteristics of the PLWD (e.g. their follow up and involvement with the memory service or dementia day programme, cognitive function, and behavioural and psychological symptoms of dementia because it could interfere with the qualitative interviewing process. However, we understand that severity of cognitive impairment and BPSD could affect the experiences of care stress.

  1. All the information provided were descriptive. This study lacks any quantitative analysis.

This is a qualitative study (line 85) and we used qualitative methods to analysis our data and present our results. Therefore, there is no quantitative analysis.

  1. There is a lack of a control group.

This is a qualitative study (line 85) and a control group is not necessary in qualitative studies.

  1. There is a lack of a table presenting the sample.

 Table 1 describes the sample and Figure 1 presents the study participant flow.

  1. The data is poorly discussed.

This comment is not specific enough for us to address. Sorry. Reviewer 1 and the Academic Editor did not express concern about our data being poorly discussed.

  1. The conclusion should reflect the findings and answer the objectives.

The aim of this study was to conduct semi-structured interviews with Chinese PLWD and/or their families to explore their understanding of dementia and experiences of living with dementia. We used qualitative methodology in this study because it lends itself more readily to explore these phenomena in-depth. We hope to use this new knowledge to develop better dementia services that are culturally appropriate and responsive to the needs of the growing number of Chinese PLWD and their families in New Zealand. (line 83-88)

We believe our conclusion (line 454-474) has reflected our findings and answered the study objectives:

Our study findings have provided some insight into ways to improve dementia care for Chinese New Zealanders. Targeted psychoeducation in the Chinese community could improve the awareness of dementia and reduce the stigma attached to dementia. This could result in earlier recognition and diagnosis of dementia and better use of healthcare and formal support services. The “brain shrinkage” term provides a mean for Chinese people to understand dementia and accept it as a medical diagnosis. We would certainly like to explore the acceptability of this term further in future research with larger groups of Chinese lay people and healthcare professionals locally and internationally. Improved access to person-centred interventions and learning about strategies for healthy ageing would provide Chinese PLWD and their families the resources to live well with dementia. Chinese family carers would benefit from structured education and support programmes to help them with negotiating the conflict between filial piety and carer stress and to instil hope while caring for their loved ones as part of the dementia journey.

Since the completion of this study, the research team has partnered with two community organizations in a new project called “Caring for people with dementia together” (“護腦同行”) with the goal of achieving better outcomes for Chinese PLWD and their families.35 Using some of our study findings, the priorities of this new project are placed on increasing dementia awareness within the Chinese community, providing psychoeducation and support for Chinese family carers, and implementing a living well programme for Chinese PLWD.

  1. Some non-English characters are including in the text. Please define clearly if it is Chinese or not. Moreover, characters with Mandarin Chinese and Cantonese Chinese were applied arbitrarily and inconsistent with the dialect spoken by the specific pseudonym of person and care partner with dementia.

Thank you for your comments. We have deleted some Chinese characters which are not critical for readers to understand our results. We included Simplified Chinese characters only in the revised manuscript (line 150-152):

We translated quotes from Chinese to English for publication purpose but illustrated some of the important quotes in Simplified Chinese characters to minimise any loss of meaning in translation.

Round 2

Reviewer 2 Report

accept